# SAVi++: Towards End-to-End Object-Centric Learning from Real-World Videos

**Gamaleldin F. Elsayed** [*][†], **Aravindh Mahendran**[*][⋄], **Sjoerd van Steenkiste**[*][⋄],
**Klaus Greff, Michael C. Mozer & Thomas Kipf**[*]
Google Research

## Abstract

The visual world can be parsimoniously characterized in terms of distinct entities with sparse interactions. Discovering this compositional structure in dynamic visual scenes has proven challenging for end-to-end computer vision approaches unless explicit instance-level supervision is provided. Slot-based models leveraging motion cues have recently shown great promise in learning to represent, segment, and track objects without direct supervision, but they still fail to scale to complex real-world multi-object videos. In an effort to bridge this gap, we take inspiration from human development and hypothesize that information about scene geometry in the form of depth signals can facilitate object-centric learning. We introduce SAVi++, an object-centric video model which is trained to predict depth signals from a slot-based video representation. By further leveraging best practices for model scaling, we are able to train SAVi++ to segment complex dynamic scenes recorded with moving cameras, containing both static and moving objects of diverse appearance on naturalistic backgrounds, without the need for segmentation supervision. Finally, we demonstrate that by using sparse depth signals obtained from LiDAR, SAVi++ is able to learn emergent object segmentation and tracking from videos in the real-world Waymo Open dataset.

Project page: `https://slot-attention-video.github.io/savi++/`

## 1 Introduction

The natural world consists of distinct entities—people, dogs, cars, trees, etc.— and its complexity emerges from the combined, mostly independent, actions of the entities. This compositional structure must be appreciated to predict future states of the world and to effect particular outcomes. People have an intrinsic understanding of objects: objects have spatiotemporal coherence, they interact when in close proximity, and they

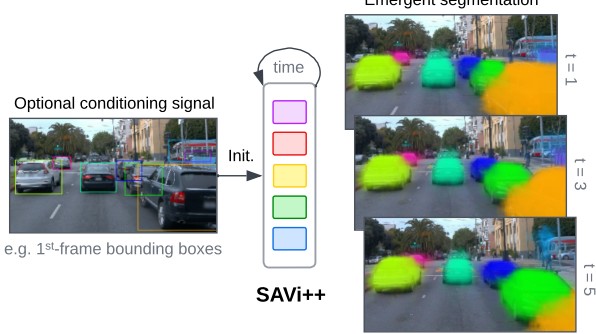

Figure 1: Emergent segmentation and tracking in SAVi++.

---

[*]Equal technical contribution. [⋄]Alphabetical order. [†]Correspondence to: `gamaleldin@google.com`
Author contributions: GFE, TK initiated and led the project. GFE, AM, SVS, TK developed the main model. GFE, AM, TK developed real-world driving data and model infrastructure. AM implemented data augmentation. GFE led ablation study. SVS led target signals analyses and metric design. GFE, AM, SVS, TK ran experiments. AM, SVS, KG, TK worked on baselines. KG, MCM provided advice at all stages and helped with project scoping. TK developed visualizations. GFE, MCM, TK worked on figure design. All authors wrote the paper.

36th Conference on Neural Information Processing Systems (NeurIPS 2022).

possess persistent, latent characteristics that determine their behavior over extended periods of time [25, 46]. Just as object-centric representations are critical to human understanding, they have the potential in machine learning to greatly improve sample efficiency, robustness, visual reasoning, and interpretability of learning algorithms [15, 35]. For example, consider the challenge faced by an autonomous vehicle operating in diverse surroundings (Figure 1). Generalization across situations requires learning about recurring entities like cars, traffic lights, and pedestrians, and the rules that govern interactions among these entities.

In human brains, the ability to organize edges and surfaces into unitary, bounded, and persisting object representations develops through experience and/or maturation from infancy and without explicit instruction via a 'core system of object representation' [46], i.e., a form of cognitive inductive bias. In deep learning, such an inductive bias has been proposed in *slot-based* architectures which segregate knowledge about individual objects into nonoverlapping but interchangeable pools of neurons. The resulting representational modularity can facilitate causal reasoning and prediction for downstream tasks [15, 44].

A grand challenge in computer vision has been to discover the compositional structure of real-world dynamic visual scenes in an unsupervised fashion. By unsupervised, we mean no segmentation information is provided that specifies which pixels belong together as part of a single object. Initial efforts focused on single-frame, synthetic RGB images [13, 14, 36, 50], but extending this work to video and more complex scenes proved challenging. A key insight to further progress was the realization that a color-intensity pixel array is not the only source of visual information readily available, at least not to human perceptual systems. The human perceptual system extracts motion and depth cues early in the processing stream [9–11, 20, 39]. These cues are correlated with object identities, and can therefore bootstrap the formation of object-centric representations [45].

The recently introduced *Slot Attention for Video* (SAVi) model [31] leveraged *optical flow* (frame-to-frame motion) as a prediction target to obtain object-centric representations of dynamic scenes involving complex 3D scanned objects and real-world backgrounds. However, motion prediction alone is insufficient to learn about the distinction between static objects and the background. Further, in real-world application domains such as self-driving cars, cameras themselves are subject to movement, which globally affects frame-to-frame motion as a prediction signal in non-trivial ways.

In the present work, we describe an enhanced slot-based model for video, referred to as *SAVi++* (Figure 2), which obtains qualitative improvements in object-centric representations by exploiting *depth* signals readily available from RGB-D cameras and LiDAR sensors. SAVi++ is the first slot-based, end-to-end trained model that successfully segments complex objects in naturalistic, real-world video sequences without using direct segmentation or tracking supervision.

A summary of our contributions is as follows:

- We introduce SAVi++: an object-centric slot-based video model that makes several key improvements to SAVi [31] by utilizing *depth prediction* and by adopting best practices for model scaling in terms of *architecture design* and *data augmentation*.
- On the multi-object video (MOVi) benchmark containing synthetic videos of high visual and dynamic complexity [16], we find that SAVi++ is able to handle videos containing complex shapes and backgrounds, and a large number of objects per scene. Improving on SAVi, our approach accommodates both static and dynamic objects and both static and moving cameras.
- Finally, we demonstrate that SAVi++ trained with sparse depth signals obtained from LiDAR enables emergent object decomposition and tracking in real-world driving videos from the Waymo Open dataset [47].

## 2   Related work

**Object-centric learning**   A growing body of research is addressing the problem of end-to-end learning of object-centric representations from raw perceptual data without direct supervision. Slot-based neural networks such as IODINE [14], MONet [4], and Slot Attention [36] rely on a factorized latent space and independent per-object decoders as inductive bias to enable object discovery in a simple auto-encoding setup. Architectures with stronger inductive biases using fixed object size, presence, or propagation priors have been explored in works such as SQAIR [33] and SCALOR [23], but generally these methods have faced challenges scaling to more complex real-world data when

relying on auto-encoding alone. Our work primarily builds on recent advances in object-centric generative models for video sequences [24, 31, 50, 52, 59]. Different from our approach, these methods have so far been unable to scale to complex real-world multi-object video data. An alternative class of methods using contrastive learning for object discovery [19, 30, 37, 56], most notably GroupViT [56] and ODIN [19], has recently achieved some success in discovering semantic groupings in real-world images. However, neither GroupViT nor ODIN model dynamics and typically fail to separate semantically similar object instances in close proximity. In our work, we follow a generative approach, but instead of tasking the decoder to generate complex visual RGB pixel data, we utilize depth information to bootstrap object-centric learning without direct supervision.

**Object discovery in driving scenes**   A range of recent methods [1, 17, 49, 53] use a multi-stage pipeline of 1) obtaining *pseudo ground truth* (PGT) segmentation or detection labels via some heuristic, and 2) training a model in a supervised fashion on PGT labels. While this class of methods achieves some success in discovering and tracking objects in real-world driving scenes, it crucially hinges on the quality of the PGT labels, requiring carefully engineered task-specific heuristics to extract objects. Earlier methods solely use clustering heuristics to extract approximate segmentation masks directly from motion trajectories for moving objects [3, 40]. In our work, we instead demonstrate that object segmentation and tracking can emerge in an end-to-end setting on complex real-world data without relying on PGT label generation.

**Cross-modal learning**   For self-supervised object-centric learning from visual data, a range of target modalities and training signals have been explored in the literature. By using motion cues from optical flow as prediction targets, several recent methods [31, 57] were able to overcome limitations of purely RGB pixel-level generative models, which frequently failed in the presence of complicated textures [27]. However, this advantage is primarily limited to discovery of moving objects. Utilizing depth targets from a simulator [2] or from sparse LiDAR [17, 49, 53], has been explored in an effort to overcome these limitations. Different from prior works utilizing multi-stage pipelines and hand-crafted heuristics for extraction of pseudo-labels from LiDAR [17, 49, 53], we directly utilize the (sparse) depth signal as target and demonstrate that this can enable emergent object segmentation and tracking on real-world driving data without any additional regularizers or pseudo-labeling techniques.

**Scaling strategies for vision models**   It is common practice to scale architectural capacity with dataset complexity and size, while making use of strong data augmentation when addressing various supervised computer vision tasks [7, 8, 18, 32]. Nonetheless, self-supervised methods for end-to-end object discovery have primarily been relying on overly simplistic and low-capacity backbone architectures [14, 36, 59], likely due to the simplicity of datasets and tasks considered in prior work. By scaling object-centric methods to larger, visually more complex datasets, we find that utilizing stronger visual backbone architectures—in combination with data augmentation—can provide substantial benefits. For simplistic datasets with lower visual complexity (and same number of examples), we found anecdotal evidence in preliminary experiments for the opposite effect: both architecture scaling and data augmentation can negatively affect object discovery performance, likely explaining why prior works have not explored these strategies.

**Depth estimation**   Recent advances in supervised monocular depth estimation (see Ming et al. [38] for a review) could be combined with our method in future work, for instance using ordinal regression losses [12], transformer architectures [43], or more complex instance-wise decoder architectures [54].

## 3   Methods

We begin by providing a brief introduction to Slot Attention for Video (SAVi), which is the starting point for our exploration. With SAVi++, we introduce several simple yet crucial improvements, which allow us to bridge the gap to complex real-world data. Our framework is summarized in Figure 2.

### 3.1   Background

Slot Attention for Video (or SAVi) is a recent state-of-the-art architecture for learning object-centric representations from video with minimal supervision. We briefly highlight some of its key components below and refer the reader for complete details to Kipf et al. [31].

SAVi can be viewed as an autoregressive encoder-decoder video model with a structured latent state composed of $K$ object slots. At a given time-step, an *encoder* first encodes the observed video frame

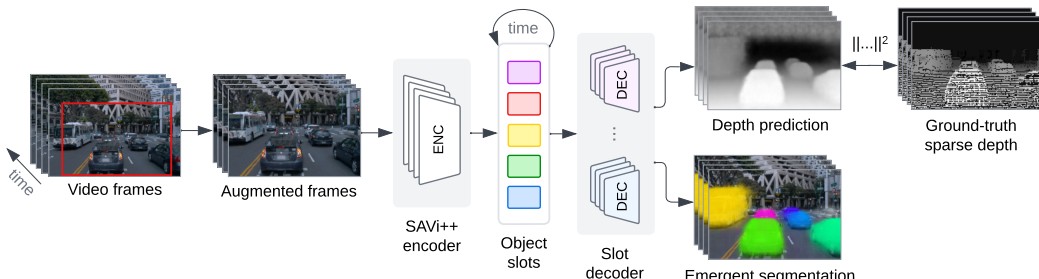

Figure 2: SAVi++ is an object-centric video model based on *Slot Attention for Video* [31], which encodes a video into a set of temporally-consistent latent variables (*object slots*). Input frames and prediction targets are augmented using random crop augmentations. Augmented frames are passed through the improved SAVi++ encoder and mapped onto object slots using an attention mechanism [36]. Slots are updated recurrently for each frame and subsequently decoded independently into a depth map and per-slot alpha masks. SAVi++ is trained using (sparse) depth targets, leading to emergence of temporally-consistent object segmentation in the decoded alpha masks.

to yield high-level image features that are useful for learning about objects. This is followed by Slot Attention [36] (the *'corrector'*), which updates the slots using these features and encourages individual slots to specialize to different parts of the observation. The content of each slot is decoded separately using a *decoder*, which additionally outputs a pixel-level alpha mask to indicate how the decoded values for each slot should be combined. Together, the mask and decoded slots determine the output of the model at the current time-step from which a loss is computed, e.g., to train the model to predict frame-to-frame motion (optical flow) for this frame. Slots for the next time-step (for the corrector to update) are obtained by applying a *predictor*, which can model interactions between slots and learn about object dynamics to predict their future state.

In addition to optical flow prediction, SAVi introduces conditioning that helps reduce uncertainty about the part-whole division into objects by pointing the model to specific locations. Indeed, in the absence of a specific downstream task, scene decomposition can be ambiguous and providing additional information as a conditioning signal may help alleviate this. The conditioning takes place via the *slot initializer*, which initializes the slots used in the initial video frame. The initialization may be learned in an unconditional setting (i.e., learn the initial slot states) or obtained by conditioning the initial state on high-level cues such as bounding boxes of objects of interest in the first video frame. This direction of attention or *input conditioning* helped SAVi to succeed in decomposing more complex visual scenes.

## 3.2 SAVi++

As SAVi relies on optical flow prediction as its main training signal for object discovery, its application is primarily limited to settings where all objects in a scene have independent motion. In addition, SAVi struggled to generalize to scenes with a moving camera, even though the optical flow field encodes information about (static) scene geometry in this case.

Here, we identify two key directions for improving SAVi and bridging its capabilities to real-world video data, while preserving its core foundation for learning object representations from video: (1) exploiting depth as a prediction signal, which is readily available in many real-world settings, and (2) utilizing model scaling strategies in terms of encoder improvements and data augmentation, which, despite being commonly used for classic vision problems, are generally underutilized for object-centric learning. Our improved approach, called SAVi++, successfully segments complex objects in naturalistic, real-world video sequences without using direct segmentation or tracking supervision.

**Exploiting depth information** Training object-centric models solely using RGB image or video frame reconstruction proves challenging in the presence of complex visual textures, frequently leading to failure modes such as clustering by color or into object-agnostic spatial regions [14, 17]. In SAVi, optical flow was proposed as a prediction signal to mitigate this issue, while still operating on visual RGB inputs [31]. However, relying solely on optical flow as a prediction target for learning about objects has a clear disadvantage: static objects, which make up the vast majority of visual entities we encounter on a daily basis, are not captured in this signal unless the observer or the entire scene is in

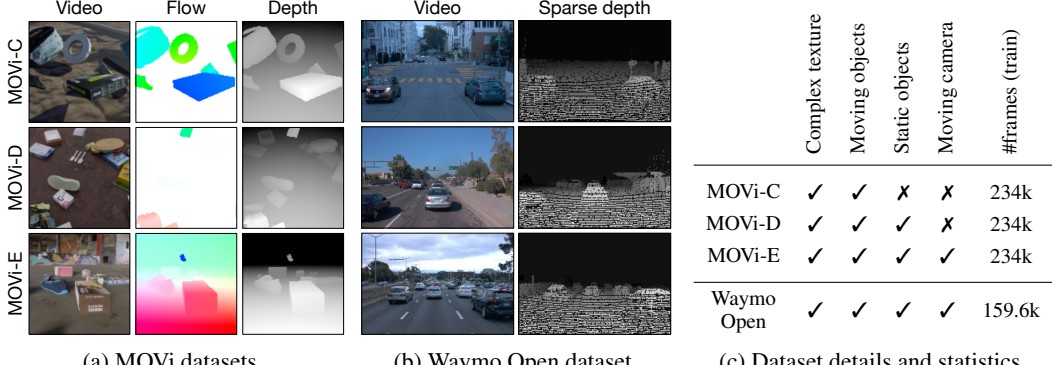

(a) MOVi datasets.   (b) Waymo Open dataset.   (c) Dataset details and statistics.

Figure 3: We consider three synthetic Multi-Object Video (MOVi) datasets [16] and the large-scale real-world driving dataset Waymo Open [47]. All datasets contain complex textures and moving objects. The MOVi datasets increase in complexity from MOVi-C (moving objects only) over MOVi-D (+static objects) to MOVi-E (+ moving cameras). Waymo Open contains all these characterisics.

motion. As a consequence, SAVi fails to represent objects that are at rest, and similarly struggles with scenes observed from a moving camera, as optical flow can prove challenging to model in this case.

Here, we explore depth as a target signal, used in conjunction with flow or even in isolation. Depth estimation has received little attention in slot-based models, yet does not suffer from the limitation of optical flow in datasets with static objects and camera movement. We thus hypothesize that depth may greatly benefit obtaining emergent object decompositions of complex videos. In terms of practical applicability, we note how depth is a readily-available signal in many real-world settings thanks to the prevalence of RGB-D cameras and LiDAR in settings like self-driving cars [47]. Even in the absence of depth sensing capabilities, this signal can be cheaply estimated from multi-camera systems [34].

In our implementation, we represent the depth signal in image space, which we encode using a log transformation $\log(1 + d)$, where $d$ is the distance of a pixel to the camera (see Figure 3a). This log-transform puts a stronger emphasis on close-by objects and—in early experiments—we found this form of normalization crucial for reliably training object-centric models using depth targets. SAVi++ is then trained to minimize the squared difference between the decoder output and this target signal. In case of multiple available targets, such as depth and flow, we concatenate the target images along the channel dimension and predict them using an otherwise unchanged model.

For sparse targets such as depth obtained from LiDAR, we ignore any points in the image space for which no signal is present in the computation of the loss. For LiDAR specifically, we obtain the x, y, z coordinates of all the LiDAR points in the self-driving car (SDC) world and compute the distance of each of the points from the LiDAR sensor. We then use the camera and LiDAR calibration parameters to project the LiDAR point distances from the SDC domain to the camera frame. This projection represents a very sparse approximation of the ground-truth depth signal (Figure 3b).

**Scaling strategies**   Visual complexity present in real-world videos necessitates a different class of encoders than those used for simple synthetic datasets. Inspired by successful visual backbone architectures for set-based supervised object detection models [6, 26], we use a more capable encoder that utilizes the ResNet34 [18] architecture followed by a transformer encoder [51] (with 4 layers, unless otherwise mentioned). To avoid computation of batch and/or temporal statistics, we replace the typical batch normalization in ResNet34 with group normalization [55]. We use a stride 1 convolution and use no max-pooling in the ResNet root block. This results in an overall backbone stride of 8 (as opposed 32), which was found to be important for retaining object decomposition capabilities. Please see the appendix for further architectural details.

Drawing inspiration from training schemes commonly used for real-world vision models [48], we further apply Inception-style cropping as data augmentation. In particular, we randomly crop a region of each frame with aspect ratio $\in [0.75, 1.33]$ such that enough of the frame is retained after cropping. The same crop is applied consistently across all frames and the resulting video is resized to the original resolution. Flow fields and depth maps are adjusted accordingly to keep them accurate and spatially aligned with the video frame.

# 4  Experiments

The goal of our experimental evaluation is twofold: 1) on synthetic video data of varying complexity we would like to analyze the potential advantages of utilizing a depth signal and model scaling strategies for learning emergent segmentation and tracking, and 2) we would like to investigate whether these improvements enable bridging the gap to complex real-world video data.

Section 4.1 covers both qualitative and quantitative comparisons of SAVi++ against baselines on the synthetic MOVi datasets. In Section 4.2, we perform an ablation study on SAVi++. Finally, in Section 4.3 we demonstrate and analyze results for a SAVi++ model applied to real-world driving videos from the Waymo Open [47] dataset.

**Datasets**  As basis for our experiments, we use videos of different scene and camera complexities (Figure 3c). We use three synthetic Multi-Object Video (MOVi) datasets (Figure 3a) introduced in Kubric [16], which are created by simulating rigid body dynamics. We narrow our investigation to MOVi datasets with complex naturalistic backgrounds and 3D-scanned everyday objects (variants C, D, and E). MOVi-C is generated using a static camera, and all objects (max. 10) are initialized to move independently. MOVi-D introduces more objects, some of which are dynamic (1-3) and the majority rests statically in the scene (10-20). Finally, MOVi-E introduces random, linear camera movement. Each video contains 24 frames sampled at 12 frames per second (fps).

We also train and evaluate SAVi++ in a real-world driving setting using the Waymo Open dataset (Figure 3b). Waymo Open is comprised of high resolution video data of $1280 \times 1920$ original resolution from a multi-camera system collected by Waymo vehicles [47]. The dataset consists of 798 train and 202 validation scenes of 20s video each, sampled at 10 fps. We subsample the dataset at 5 fps both for training and validation. The dataset also includes LiDAR signals that we use to compute sparse depth maps as discussed in Section 3.

**Training setup**  For all our experiments, unless stated otherwise, we resize frames to a height of 128 pixels while keeping the aspect ratio fixed, resulting in a $128 \times 128$ resolution for MOVi datasets, and a resolution of $128 \times 192$ for Waymo Open. We train SAVi++ for 500k steps on Tensor Processing Unit (TPU) accelerators with a batch size of 64 using Adam [29].

We train on randomly sampled sub-sequences of only 6 frames using 24 slots for MOVi and 11 slots for Waymo Open. See appendix for further training details and hyperparameters.

## 4.1  SAVi++ improves object-centric learning on complex synthetic video data

We investigate whether the key changes introduced to SAVi [31], which constitute our improved SAVi++ model, allow us to overcome limitations of SAVi and address the most challenging synthetic multi-object video (MOVi) benchmarks introduced in Kubric [16].

**Setup**  We train all models independently on each dataset variant. Both SAVi and SAVi++ are trained in a conditional setting where we initialize slots using ground-truth bounding box information in the first frame. We report the same segmentation metrics as in prior work, i.e. Foreground Adjusted Rand Index (FG-ARI) [21, 42] and Mean Intersection over Union (mIoU). FG-ARI is a permutation-invariant clustering similarity metric frequently used for evaluating scene decomposition quality. It compares discovered segmentation masks with ground-truth masks while ignoring any pixels that belong to the background. It is sensitive to temporal consistency of masks, but insensitive to their ordering. The mIoU metric is a standard segmentation metric, here adapted for video as in [5]. We note that this implementation is sensitive to the correct ordering of masks, i.e. it also measures whether models used the conditioning signal (here, first-frame bounding boxes) correctly.

**Baselines**  Besides comparing to SAVi [31], the most representative prior method for the task we are interested in, we compare against a range of baselines aimed at establishing the difficulty of the unsupervised, bounding box-conditioned video object segmentation task: 1) a bounding box copy (*BBox copy*) baseline, which simply repeats the first-frame boxes throughout the video, 2) a learned *BBox propagation baseline* that does not receive visual inputs, to test for easily exploitable biases in the datasets, 3) k-Means clustering baselines, that cluster the flow and/or depth signal across the video sequence (initialized using the ground-truth object centers in the first frame), and 4) a label propagation baseline, that uses visual features to propagate the initial boxes (rendered as rectangular masks) across the video, based on Contrastive Random Walks (CRW) [22]. See appendix for further details.

Table 1: MOVi results in terms of mean score $\pm$ standard error (5 seeds) from evaluating SAVi++ and baseline models on validation set video sequences of increased length (24 frames). *: we use the official implementation of CRW [22], which does not report FG-ARI.

| | mIoU↑ (%) | | | FG-ARI↑ (%) | | |
| Model | MOVi-C | MOVi-D | MOVi-E | MOVi-C | MOVi-D | MOVi-E |
|---|---|---|---|---|---|---|
| BBox copy | 12.3 | 42.8 | 32.9 | 11.8 | 68.0 | 54.7 |
| BBox propagation | $22.9_{\pm 0.1}$ | $26.7_{\pm 0.8}$ | $24.1_{\pm 1.1}$ | $9.6_{\pm 0.5}$ | $24.9_{\pm 3.7}$ | $18.4_{\pm 3.9}$ |
| K-Means (depth) | $7.1_{\pm 0.3}$ | $6.0_{\pm 0.4}$ | $5.4_{\pm 0.3}$ | $26.3_{\pm 1.0}$ | $30.9_{\pm 0.7}$ | $32.2_{\pm 0.6}$ |
| K-Means (flow) | $10.7_{\pm 0.5}$ | $7.4_{\pm 0.4}$ | $6.0_{\pm 0.3}$ | $26.5_{\pm 1.0}$ | $30.9_{\pm 0.8}$ | $33.1_{\pm 0.7}$ |
| K-Means (flow+depth) | $10.6_{\pm 0.6}$ | $6.7_{\pm 0.4}$ | $5.3_{\pm 0.3}$ | $26.6_{\pm 1.0}$ | $35.9_{\pm 1.0}$ | $34.8_{\pm 0.7}$ |
| CRW [22] | $27.8_{\pm 0.2}$ | $45.3_{\pm 0.0}$ | $\mathbf{47.5}_{\pm 0.1}$ | * | * | * |
| SAVi [31] | $43.1_{\pm 0.7}$ | $22.7_{\pm 7.5}$ | $30.7_{\pm 4.9}$ | $77.6_{\pm 0.7}$ | $59.6_{\pm 6.7}$ | $55.3_{\pm 5.8}$ |
| SAVi++ (ours) | $\mathbf{45.2}_{\pm 0.1}$ | $\mathbf{48.3}_{\pm 0.5}$ | $47.1_{\pm 1.3}$ | $\mathbf{81.9}_{\pm 0.2}$ | $\mathbf{86.0}_{\pm 0.3}$ | $\mathbf{84.1}_{\pm 0.9}$ |

**Results** Quantitative results can be seen in Table 1 and qualitative results on MOVi-E in Figure 4a. The *BBox copy* method serves as a trivial baseline which a learning-based approach should outperform. While the original SAVi model does so on MOVi-C, it clearly fails to model the more complex MOVi-D and -E datasets. The BBox copy baselines is—perhaps unsurprisingly—strongest on MOVi-D, where most objects are static. SAVi++ outperforms this baseline on all datasets, indicating that it learns non-trivial segmentation and tracking capabilities. Indeed, this advantage does not solely come from fitting certain biases in the datasets, as a learned BBox propagation baseline (using the same predictor as in SAVi++) that does not receive visual input, fails to generalize to unseen evaluation videos. It is worth noting that neither of the MOVi tasks can easily be solved by simply clustering the target signals, as the results for the k-Means baselines demonstrate.

Compared to CRW it can be seen how SAVi++ yields markedly better mIoU on MOVi-C and D, while performance on MOVi-E is similar. Note that, unlike SAVi++, CRW is merely capable of propagating pixel-level annotations across frames in a video and does not by itself produce instance-level object segmentations or corresponding object-representations that could be used for down-stream tasks. Finally, comparing SAVi++ and SAVi directly, we see that SAVi++ overcomes the primary limitations of SAVi on the harder MOVi-D and -E datasets, both quantatively (Table 1) and qualitatively (Figure 4a), while also improving performance on MOVi-C.

**Discussion** It is evident that a small number of critical changes to SAVi [31], namely utilizing depth targets, a stronger architecture, and data augmentation, can have dramatic consequences on the ability of this slot-based model to learn emergent object segmentation and tracking in complex video sequences. The difference between SAVi++ and SAVi is especially evident for the more complex datasets in our study (e.g., improving the mIoU score on MOVi-E from 30.7% to 47.1%; see also Figure 4a). These results demonstrate that SAVi++ is better suited for various data complexities in terms of object dynamics and camera movement, which are likely to exist in real-world data.

### 4.2 Ablation study

In this section, we report results of an ablation study to gauge the contribution of the different components of SAVi++. The three main ingredients of SAVi++ are 1) the use of depth as training target, 2) the extra capacity added to SAVi by including a transformer encoder, and 3) the use of data augmentation. Figure 4b shows a systematic ablation of each of those components. Removing the transformer encoder reduces object segmentation quality, yet the degradation in performance is relatively limited. While data augmentation only has a mild effect on the simpler MOVi-C dataset, it makes a substantial difference on the more challenging datasets, MOVi-D and E. Finally, removing depth targets reduces performance further and is particularly catastrophic on MOVi-E.

In fact, we find that training *solely* using depth targets without relying on predicting optical flow as well (see *w/o Flow* in Figure 4b) still allows the model to accurately segment and track objects, especially on the more complex MOVi-D and E datasets. This result is particularly strong on MOVi-E where jointly predicting optical flow presents a difficult task for scenes with camera movement. Further, training on depth targets was very crucial to obtain good performance on the most complex

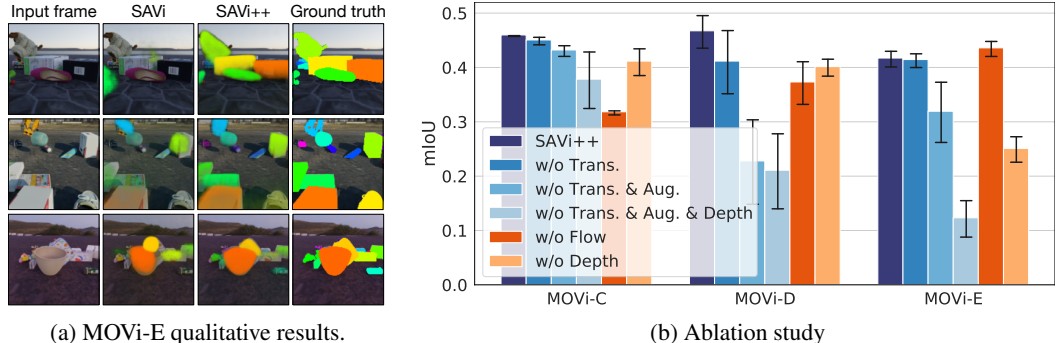

| Input frame | SAVi | SAVi++ | Ground truth |

(a) MOVi-E qualitative results.

(b) Ablation study

Figure 4: **Left**: Qualitative results of SAVi++ compared to SAVi [31] on the synthetic MOVi-E dataset with camera motion. **Right**: SAVi++ ablation study on MOVi-C, D, and E. Bars reflect validation set mIoU (mean ± standard error for 5 seeds). We ablate: 1) the transformer encoder (*w/o Trans.*), 2) data augmentation (*w/o Trans. & Aug.*), and 3) depth targets (*w/o Trans. & Aug. & Depth*). We further report results for training without flow, while only using depth targets (*w/o Flow*).

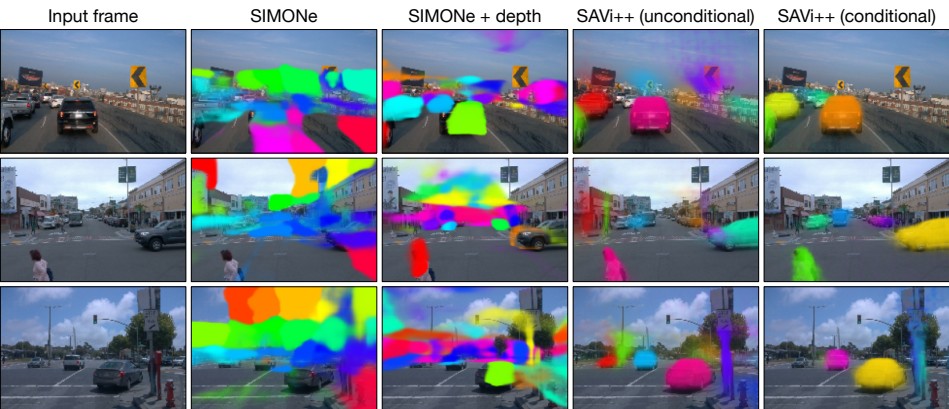

| Input frame | SIMONe | SIMONe + depth | SAVi++ (unconditional) | SAVi++ (conditional) |

Figure 5: Qualitative segmentation comparison on the Waymo Open validation set. Naive application of a SIMONe [24] baseline model to this dataset results in failure, while adapting SIMONe to predict (sparse) depth maps yields rough (but frequently misaligned) segmentation masks. SAVi++ generally produces highly accurate segmentation masks, while its unconditional results are promising. Here, we hide masks that occupy more than 1300 pixels on average per frame to ease interpretability.

synthetic data MOVi-E as demonstrated with the large drop in mIoU when ablating depth and relying only on optical flow to train the model.

### 4.3 SAVi++ enables emergent segmentation on real-world driving data

In the previous section, we found that solely using depth as a training target can be sufficient to learn emergent object segmentation and tracking. This finding provides a strong motivation for scaling this class of methods to real-world data, where the availability of optical flow relies on approximate and potentially inaccurate flow estimation methods, whereas depth can be accurately measured using technologies like LiDAR. To investigate this possibility, we use the Waymo Open dataset [47], which includes videos obtained from cameras mounted on cars in various traffic environments.

**Setup** To obtain a depth signal, we project 3D LiDAR points into the camera frame, resulting in a very sparse depth image for each time step (see Figure 3b for examples). We exclude pixels that do not have a valid LiDAR point when computing the L2 loss in image space. We train SAVi++ with 11 slots on 6 frames and evaluate the model on sequences of 10 frames. Due to the absence of ground-truth segmentation labels in Waymo Open, we quantitatively measure performance compared to ground-truth bounding boxes using three metrics. The Center-of-Mass (CoM) distance measures the average Euclidean distance between the centroid of the predicted segmentation masks and the centers of the ground-truth bounding boxes. We report the centroid distance normalized by the maximum achievable distance in the video frame. Additionally, we separately measure the fraction

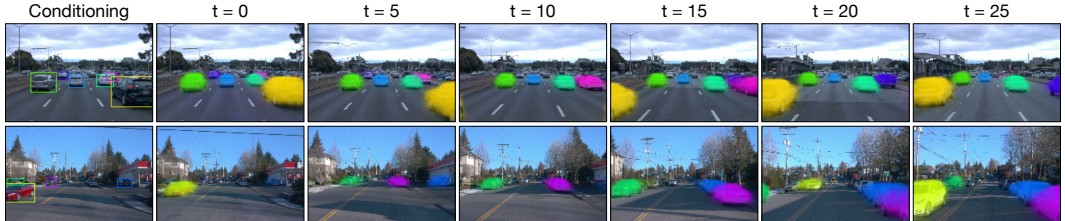

| Conditioning | t = 0 | t = 5 | t = 10 | t = 15 | t = 20 | t = 25 |

Figure 6: Waymo Open qualitative results of SAVi++ (conditional) over long sequences.

of cases where any sort of segment is predicted when a valid ground-truth box exists, denoted as bounding box recall (B. Recall). The Bounding Box mIoU (B. mIoU) is analog to mIoU using predicted and ground-truth bounding boxes. The former are obtained by training a readout MLP to predict bounding boxes from the slot representations. See appendix for further details.

**Baselines**   We quantitatively compare to the subset of previous baselines that work with (sparse) depth. Further, we report qualitative results for SAVi++ in the unconditional setting, i.e. without providing first-frame bounding boxes to the model to initialize slots, and compare to SIMONe [24] as a representative object-centric video model baseline from the literature.

**Results**   Quantitative results can be seen in Table 2 and qualitative results in Figures 5–6. We find that SAVi++ markedly outperforms the BBox copy and propagation baselines, as well as the clustering baseline in terms of object tracking. Further, the bounding box recall is high indicating that valid objects are rarely ignored. The qualitative results in Figures 5–6 even better reflect the significance of SAVi++'s performance as well as its potential utility for object-centric representation learning from real-world videos (for SAVi++ results divided per object category see appendix).

Our results using sparse depth targets suggest that SAVi++ does not need complete (i.e. dense) depth supervision. To investigate how *accurate* this signal needs to be, we explored the degree of sensitivity of SAVi++ to noise in the depth signal. We trained SAVi++ with noisy depth targets by applying additive Gaussian noise to the ground-truth sparse LiDAR depth signals with standard deviations of 10cm, 20cm and 40cm. We found that SAVi++ was able to retain its emergent tracking performance even at the highest considered noise scale of 40cm (see Table 3 in appendix).

We additionally experimented with removing the bounding box conditioning in SAVi++ in the initial frame. Removing this conditioning signal and using a learned initialization together with a simplified encoder also yielded good object decompositions (see SAVi++ (unconditional) in Figure 5). Compared to using plain SIMONe [24], we observe that SAVi++ (unconditional) performs markedly better. Interestingly, modifying the non-autoregressive SIMONe baseline similar to SAVi++ by predicting sparse depth instead of RGB also showed improvement in object emergence. This gives further evidence that using depth is suitable for learning object-centric representations from real videos. Quantitatively, SAVi++ achieves a CoM distance of $6.9 \pm 0.5$ while SIMONe (with depth loss)

Table 2: Waymo Open results (mean $\pm$ standard error in %, 3 seeds) from evaluating models on sequences of 10 frames. SAVi++ HR is a variant trained on higher-resolution ($256 \times 384$) video frames.

| Model | CoM↓ | (%) B. mIoU↑ | B. Recall ↑ |
|---|---|---|---|
| BBox Copy | 5.0 | 44.3 | 100 |
| BBox Prop. | $5.1 \pm 0.1$ | $38.5 \pm 0.5$ | 100 |
| K-Means (depth) | $13.0 \pm 0.1$ | – | 100 |
| SAVi (RGB) | $21.5 \pm 1.8$ | $7.9 \pm 0.9$ | $95.8 \pm 2.7$ |
| SAVi (depth) | $24.7 \pm 0.7$ | $10.3 \pm 2.4$ | $97.4 \pm 0.6$ |
| SAVi++ | $\mathbf{4.4} \pm 0.2$ | $\mathbf{49.7} \pm 0.7$ | $96.5 \pm 0.7$ |
| SAVi++ HR | $\mathbf{3.9} \pm 0.1$ | $\mathbf{51.9} \pm 0.4$ | $96.2 \pm 0.4$ |
| Supervised | $1.1 \pm 0.0$ | $67.6 \pm 0.6$ | |

achieves $7.4 \pm 0.2$[1] over a sequence of 12 frames at test time, evaluated using Hungarian matching.

We show qualitative results for longer sequences in Figure 6 and in video-format in the supplementary material. It is worth noting that SAVi++ was only trained on 6 frames and did not receive any tracking supervision. Interestingly, we find that objects are often consistently tracked until the moment they leave the scene. At this stage, slots are freed up again and tend to bind to previously unexplained or new objects. This behaviour indicates that our reported tracking metrics are an underestimation of the capabilities of the model, as such re-binding is not accounted for. It is, however, conceivable

---

[1]These baseline results are improved compared to an earlier version of the paper by using exactly the same depth target transformation as for SAVi++.

that re-binding events could be identified post-hoc if one were to use the representations learned by SAVi++ for downstream tasks, which is an interesting avenue for future work.

### 4.4 Limitations

With SAVi++, we demonstrated the first proof of concept that an emergent object-centric decomposition of real-world complex videos is possible with an end-to-end slot-based approach. Yet, there is still a lot of room for improvement.

**Reliance on conditioning**   We focused our exploration on the conditional setup where we provided cues in the form of bounding boxes of objects in the first frame. Although the use of such "object hints" may share some similarity to how human visual attention (and how humans parse a visual scene) can be directed via external signals (e.g., via gestures such as pointing), it ultimately limits the practical applicability of our approach. Preliminary results with unconditional SAVi++ suggest that this information may not be strictly necessary and could be removed in future research.

**Reliance on ground-truth target signals**   In a similar vein, the reliance of SAVi++ on ground-truth target signals for training is a limitation that may affect its practical applicability. Fortunately, LiDAR sensors for depth estimation are readily available in many application domains (such as in robotics and self-driving), and there is also a rich literature on monocular depth estimation. While estimated, depth (or flow) are expected to be noisier compared to the signals considered in our experiments, our experiment with "noisy depth" offers an initial sign that this may not affect performance much.

**Gap to videos recorded in the wild**   It is also important to point out that although Waymo Open offers a challenging real-world benchmark for learning about objects, its videos are relatively structured compared to real-world videos recorded "in the wild", and especially heavy on cars, roads, traffic signs, pedestrians, etc. Other datasets, such as DAVIS [41] or Kinetics [28] offer greater complexity in that regard and it is foreseeable that further development of SAVi++ will be needed to truly support these. An example of this is that objects in Waymo Open usually do not re-appear, which is an aspect that is currently not explicitly modeled in SAVi++ (e.g. to ensure that the same object is re-captured by the same slot). More generally, there is substantial headroom to improve the modeling of disappearing and reappearing objects in future work, such as by explicitly modeling object presence [33], or by explicitly attending to past latent states [58].

**Gap to supervised approaches**   Finally, we note how both in the conditional and the unconditional setting, the segmentation and tracking performance, though impressive given the minimal amount of supervision the model receives, still qualitatively lags behind supervised approaches. Improving on the temporal consistency of object tracks, especially in the unconditional setting, is another promising direction for future work.

## 5   Conclusion

We demonstrate that object tracking and segmentation can emerge from utilizing information about scene geometry in the form of depth signals in complex video data with slot-based neural architectures. We utilize a series of synthetic multi-object video benchmarks with increasing complexity to find a simple yet effective set of changes to an existing state-of-the-art object-centric video model (SAVi), allowing us to bridge the gap from synthetic to complex real-world driving videos.

Our work marks a first step towards building end-to-end trainable systems that learn to perceive the world in an object-centric, decomposed fashion without relying on detailed human supervision. While many open challenges remain, this result evidences that object-centric deep neural networks are not inherently limited to simple synthetic environments, and we are excited about the potential for this class of methods to radically reduce the need for human supervision in building scalable perceptual systems for the real world.

## 6   Acknowledgements

We would like to thank Ben Caine, Alex Bewley and Pei Sun for assistance with self-driving data. We are grateful to Jie Tan, Daniel Keysers, David Fleet, Matthias Minderer, Mehdi Sajjadi and Mario Lučić for general advice and feedback.

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
