# OpenReview forum: "SAVi++: Towards End-to-End Object-Centric Learning from Real-World Videos"
_NeurIPS.cc/2022/Conference — NeurIPS 2022 Accept_

### Official Review · Reviewer_GFoH · 2022-07-11

**Rating:** 3
**Confidence:** 3
**Soundness:** 2 fair
**Presentation:** 3 good
**Contribution:** 2 fair

**Summary:**

Brief Summary: The paper proposes an extension of a previous work SAVI [30] which used optical flows as a training signal to further include depth maps obtained from RGB-D cameras / LiDAR. The main goal is to learn from videos which have depth-channels but no instance/semantic segmentation ground-truths.

Experiments on MOVi dataset types C,D, and E which are created synthetically, as well as on Waymo dataset obtained from real world data, show their proposed method SAVI++ outperforms compared baselines.

**Questions:**

Q1. How important is the high resolution in waymo dataset?

Q2. What is the main bottleneck in training time, given that the underlying datasets are not very large? Is it sequential time-steps for filling in the object slots?

Q3. In L100, it is noted that no PGT data is needed. Does using PGT data give improvements, and if so how much?

**Limitations:**

Yes. Some more discussions on where the models are likely to fail such as low-data domain, as evidenced by L25 of suppl, could be useful.

**Strengths And Weaknesses:**

Pros:

1. With ever increasing 3d input data information, it is imperative that models are able to utilize such information for semantic segmentation tasks. Further, 3d data enables learning about static objects. As such, there is a clear motivation for the problem at hand.

2. The paper provides good visualizations such as Fig3, Fig5, as well those in supplementary material.

Cons:

1. In my opinion, the paper makes too general claims without providing previous citations while their actual technical contributions and evidence is far more limited. For instance, L48 suggests extending previous work is more challenging but doesn't explain why. L63 suggests it is the first model for segmentation without direct supervision or tracking supervision. This is strictly not true, given literature in domain adaption where one trains on synthetic segmentation dataset (such as Synthia or GTA) and domain adaptation transfer to Cityscapes (see [Ref1] for an example). Similar assertion is made in L3, but this ignores literature in video-text pretraining (see [Ref2] as example).

2. On novelty: the technical novelty in the paper is slightly incremental. given that it mainly suggests using 3d-depth map instead of flow maps used in previous work.

3. On Model architecture:

(i) The reason for using Resnet34 is unclear (L193). What metric is being used to decide this ("more capable encoder" is too vague).

(ii) Why not use something like Vision-Transformer instead, and remove the additional transformer encoder? Design-wise that looks much simpler?

4. On Experiments:

(i) MOVI is synthetic dataset, and Waymo dataset appears to be quite small with only 200 validation videos. More datasets should be tried such as Cityscapes, Cityscapes-VPS, VIPER (see [Ref3]).

(ii) The proposed method performs significantly worse on MOVI-A,B (as shown in Table 1 of supplementary material).  If the reason is indeed due to overfitting on simpler domains as claimed in suppl L26, there should be some experiment with same encoder.

(iii) The compared baselines in Table 1. are quite weak. The authors should compare their model with other methods trained on depth maps, at least some naive extensions of other previous works to use 3d-maps (such as for ODIN [19]).

(iv) The authors should show fine-tune performance on other segmentation tasks such as coco and pascal.

(v) Some design decisions on the choice of visual encoders should be empirically justified.

[Ref1]: Hoyer, Lukas, Dengxin Dai, and Luc Van Gool. "HRDA: Context-Aware High-Resolution Domain-Adaptive Semantic Segmentation." arXiv preprint arXiv:2204.13132 (2022).

[Ref2]: Miech, A., Alayrac, J. B., Smaira, L., Laptev, I., Sivic, J., & Zisserman, A. (2020). End-to-end learning of visual representations from uncurated instructional videos. In Proceedings of the IEEE/CVF Conference on Computer Vision and Pattern Recognition (pp. 9879-9889).

[Ref3]: Kim, Dahun, Sanghyun Woo, Joon-Young Lee, and In So Kweon. "Video panoptic segmentation." In Proceedings of the IEEE/CVF Conference on Computer Vision and Pattern Recognition, pp. 9859-9868. 2020.

---

> ### Author Response · Authors · 2022-08-02
> **Response to Reviewer GFoH**
>
> Thank you for the review and comments on our paper. We understand that the reviewer is on the fence about the significance of our contribution and evaluation and we hope the detailed replies below and additional experiments summarized in the general response above can alleviate some of these concerns.
>
> **“L48 suggests extending previous work is more challenging but doesn't explain why.”**
>
> L47-48 states that “Initial efforts focused on single-frame, synthetic RGB images [13, 14, 35, 46], but extending this work to video and more complex scenes proved challenging.”, which is a slightly different point. We do not intend to claim that extending one work is more challenging than another, although the case for extending SAVi is straightforward: it is the SOTA end-to-end unsupervised learning approach to learning about objects in videos. Moreover, aside from the slot-based representations and competition among slots, it consists of a fairly standard architecture, which makes it a good starting point.
>
>
> **“L63 suggests it is the first model for segmentation without direct supervision or tracking supervision. This is strictly not true, given literature in domain adaption where one trains on synthetic segmentation dataset (such as Synthia or GTA) and domain adaptation transfer to Cityscapes (see [Ref1] for an example).”**
>
> L61-63 states that “SAVi++ is the first slot based, end-to-end trained model that successfully segments complex objects in naturalistic, real-world video sequences without using direct segmentation or tracking supervision.”. To the best of our knowledge this is an accurate statement. Notably, it excludes multi-stages approaches that do (supervised) segmentation pre-training, transfer learning, or domain adaptation, which is not the focus of this work. We would also like to emphasize that we are concerned with end-to-end unsupervised instance segmentation as opposed to supervised segmentation, which appears to be the focus of Ref1.
>
> **“Similar assertion is made in L3, but this ignores literature in video-text pretraining (see [Ref2] as example).”**
>
> L3 states that “Discovering this compositional structure in dynamic visual scenes has proven challenging for end-to-end computer vision approaches unless explicit instance-level supervision is provided.”, which is supported by numerous works. Note that we do not make any claims about the feasibility of (video-text) pretraining, transfer learning and domain adaptation in the unsupervised case and specifically focus on end-to-end approaches as considered here.
>
> **“The reason for using Resnet34 is unclear (L193). What metric is being used to decide this ("more capable encoder" is too vague).”**
>
> Thank you for pointing this out. In our preliminary experiments we have considered a simpler CNN architecture (i.e. the default in the SAVi paper), the ResNet34 model (which was found to lead to better performance in the SAVi paper on their datasets) and a ResNet18 model (to strike a middle ground between these). On MOVi-C/D/E the ResNet34 performed best overall.
> We will clarify how we arrived at this decision since we agree that “more capable encoder” is too vague.
>
> **“Why not use something like Vision-Transformer instead, and remove the additional transformer encoder? Design-wise that looks much simpler?”**
>
> Thank you for pointing this out. Our intuition was to stay in line with recent state-of-the-art instance/panoptic segmentation models, a large majority of which use hybrid CNN+Transformer encoders, see Mask2Former (Cheng et al., CVPR 2022) and k-means Mask Transformer (Yu et al., ECCV 2022) for recent SOTA examples.
>
> **“MOVI is synthetic dataset, and Waymo dataset appears to be quite small with only 200 validation videos.More datasets should be tried such as Cityscapes, Cityscapes-VPS, VIPER (see [Ref3]).”**
>
> We agree that the paper would be even stronger if SAVi++ could be shown to work on other real-world datasets, especially those having different characteristics such as VIPER. However, the diversity of Waymo Open videos should also not be underestimated as it concerns scenes recorded outside in the open that include lots of variation in scene backgrounds, clutter, and camera motion, unlike what is encountered in indoor scenes.
>
> More importantly, prior object-decomposition approaches are mainly limited to synthetic datasets, which offer limited application domains. In that sense our results on Waymo Open demonstrate a true step toward end-to-end object-centric learning from real-world video, despite its size and object variability.

---

> > ### Author Response · Authors · 2022-08-02
> > **Response to Reviewer GFoH (continued)**
> >
> > **“The proposed method performs significantly worse on MOVI-A,B (as shown in Table 1 of supplementary material). If the reason is indeed due to overfitting on simpler domains as claimed in suppl L26, there should be some experiment with same encoder.”**
> >
> > Thank you for pointing this out. Indeed, our preliminary experiments indicated that this is the case, i.e. we observe overfitting on these datasets in the depth prediction setting, and using a simpler CNN encoder alleviates this. With MOVi-A/B being limited to simple geometric objects and our goal of scaling to real-world videos, we did not give this much further consideration. However, we agree that it could be useful to add this and will include this comparison.
> >
> > **“The compared baselines in Table 1. are quite weak. The authors should compare their model with other methods trained on depth maps, at least some naive extensions of other previous works to use 3d-maps (such as for ODIN [19]).”**
> >
> > The baselines in Table 1 include SAVi and CRW, both of which are representative and strong baselines for the setting we consider. For example, CRW achieves strong performance on MOVI-E.
> >
> >
> > It would be interesting to adapt a method like ODIN to 1) produce more instance-centric segmentations (as opposed to segmentations that primarily focus on semantic aspects), and 2) to a (conditional) video setting. We believe that both of these points are out of scope for our work, but would be interesting for future papers to explore.
> >
> > **“The authors should show fine-tune performance on other segmentation tasks such as coco and pascal.”**
> >
> > As we highlighted previously, the focus of this work is not on transfer learning or domain adaptation, hence, while we believe this type of analysis is an interesting future direction, it is out of scope for this work.
> >
> > **“How important is the high resolution in waymo dataset?”**
> >
> > We do not find this to be very important: A resolution of 128x192 was sufficient to obtain good performance for SAVi++ on Waymo. In the interest of increasing the resolution further (e.g., as may be required for some future dataset) we have added one experiment using a higher resolution of 256 x 384 (i.e., SAVi HR). Our results show that the increased resolution improves performance slightly but is not essential.
> >
> > **“What is the main bottleneck in training time, given that the underlying datasets are not very large? Is it sequential time-steps for filling in the object slots?”**
> >
> > In general we do not find that the current training time is prohibitive for any of the datasets we explore. Indeed, we can run on Waymo Open (one of the largest available labeled driving datasets with high quality LiDAR). More generally, the training time is mainly determined by the number of time-steps, and the number of object slots.
> >
> > **“it is noted that no PGT data is needed. Does using PGT data give improvements, and if so how much?”**
> >
> > That is an interesting question for future work. As part of the rebuttal we have conducted an experiment with a supervised version of SAVi++ w/ actual GT (similar to TrackFormer), which resulted in 1.2 ± 0.0 % CoM and 68 ± 0.3 % BBox mIoU (for two seeds) compared to 4.4 ± 0.0 % and 50.5 ± 0.3 % for SAVi++. We expect these supervised numbers to be an upper bound for what amount of improvement is possible using PGT.
> >
> > **“Some more discussions on where the models are likely to fail such as low-data domain, as evidenced by L25 of suppl, could be useful.”**
> >
> > Thank you for this suggestion. We will incorporate a discussion on this as part of the revised limitation section.

---

> > ### Comment · Reviewer_GFoH · 2022-08-09
> > **Response to authors**
> >
> > I have read the authors rebuttal as well as the additional experiments. I thank the authors for the detailed response.
> >
> > Unfortunately, central issue of comparison against weak baselines is not resolved.
> >
> > In particular, answers to Cons 4. (i)-(v) remain unsatisfactory to me. Dataset choice is not strong, and more segmentation datasets with more existing baselines should be shown. Results on MOVI-A,B is in fact poorer than previous method.
> >
> > Most importantly, the compared baselines are very weak; they don't even take into account 3d data. The main takeaway from the paper seems to be additional signals boost performance, which is interesting, but not particularly unexpected. Essentially, the paper is comparing apples to oranges; proposed method has access to additional data while baselines don't. This is why I suggested some trivial extension of ODIN to use 3d-maps. I don't believe this is outside the scope of the paper, instead integral to judging the paper.
> >
> > Basically, the authors need to show that the proposed method of using 3d-maps is in fact the best way, and no other simpler methods can work. If a direct late-fusion kind of technique gives comparable performance, then the proposed method has very little relevance. More data points are needed to adequately qualify the effectiveness of the method.
> >
> > At this point, I would keep my score of 3, primarily because the experiments are not conclusive of the effectiveness of the proposed method.

---

### Official Review · Reviewer_uKMC · 2022-07-11

**Rating:** 6
**Confidence:** 4
**Soundness:** 3 good
**Presentation:** 3 good
**Contribution:** 3 good

**Summary:**

Learning the compositional structure (e.g. scene graph) in dynamic visual scenes without labels (e.g. object masks) has proven challenging.  Slot-based models leveraging motion cues have recently made progress in learning to represent, segment, and track objects without direct supervision. However, these methods only work on synthetic data and fails on complex real-world multi-object videos. This paper proposes to exploit depth signals to improve object-centric learning. It introduces SAVi++, an object-centric video model which is trained to predict depth signals from a slot-based video representation. The paper claims that by using sparse depth signals obtained from LiDAR, SAVi++ is able to learn emergent object segmentation and tracking from videos in the real-world Waymo Open dataset.

**Questions:**

For Waymo dataset, the authors should compare with SOTA self-supervised instance segmentation methods, e.g. SAVi and GroupViT. The paper should also mention the performance of SOTA supervised methods to clarify on the gaps between these two type of methods.

Can SAVi++'s depth prediction module benefit from the large body of monocular depth estimation paper?

It seems that the feature learning can use a strong baseline, e.g.  Self-Supervised Representation Learning from Flow Equivariance, ICCV 2021.

**Limitations:**

The paper does not seem to have stated its limitations. The work has several major limitations, e.g. relying on depth information, far from being competitive with SOTA supervised models, and lacking experimental results on real-world videos.

**Strengths And Weaknesses:**

Strengths

SAVi++ improves on SAVi with predicting depth signals, and utilizing stronger visual backbone architectures in combination with data augmentation.

SAVi++ evaluates on Waymo Open dataset.

Weaknesses

Despite the claims "End-to-End Object-Centric Learning from Real-World Videos", SAVi++ has very limited results on only one real world video dataset, Waymo. The results on Table 2 do not compare with any significant prior work. For more convincing experimental results, the authors are encouraged to consider indoor datasets as well, SUN RGB-D.

The paper ignores a large body of work on monocular depth estimation, e.g. Monocular Depth Estimation Using Relative Depth Maps, CVPR, 2019.

---

> ### Author Response · Authors · 2022-08-02
> **Response to Reviewer uKMC**
>
> Thank you for the careful review and useful feedback. We provide a point-by-point response to the remaining comments and questions below.
>
> **“Despite the claims "End-to-End Object-Centric Learning from Real-World Videos", SAVi++ has very limited results on only one real world video dataset, Waymo. The results on Table 2 do not compare with any significant prior work. For more convincing experimental results, the authors are encouraged to consider indoor datasets as well, SUN RGB-D.”**
>
> Thank you for pointing this out. Compared to real-world videos in the wild, Waymo Open is indeed relatively structured and certainly heavy on cars. Other datasets, such as SUN RGB-D  offer greater complexity in that regard and it is foreseeable that further development of SAVi++ will be needed to truly support these. We will update the limitations section in the revised version of the paper to reflect this. Having said that, the diversity of Waymo Open videos should also not be underestimated as it concerns scenes recorded outside in the open that include lots of variation in scene backgrounds, clutter, and camera motion, unlike what is encountered in indoor scenes. We also note how prior object-decomposition approaches are mainly limited to synthetic datasets, which offer limited application domains. In that sense we argue that our results on Waymo Open are a true step *toward* end-to-end object-centric learning from real-world videos as the title indicates.
>
> Regarding the results in Table 2, we have added a quantitative comparison to two variants of the SAVi model (trained with RGB or Depth targets), which provides a better indication of the improvement of SAVi++. Our results below show how SAVi++ compares to these baselines.
>
> |Model            |CoM(%)     | B. mIoU(%)|
> |-----------------|-----------|-----------|
> |SAVi (RGB)       |21.5 ± 1.8 |7.9 ± 0.9  |
> |SAVi (Depth)     |17.5 ± 5.4 |21.7 ± 8.2 |
> |SAVi++           |4.4 ± 0.0  |50.5 ± 0.3 |
>
>
> **“The paper ignores a large body of work on monocular depth estimation, e.g. Monocular Depth Estimation Using Relative Depth Maps, CVPR, 2019.”**
>
> Thank you for bringing this body of related work to our attention, we will update our discussion of related work to incorporate this. More generally, advances in monocular depth estimation architectures and losses only strengthen our work by providing alternative routes through which depth signals can be obtained. Here we opted for a simple depth estimation loss & architecture to stay closely to the SAVi model and preserve the ability to decompose video scenes into objects. For future work, we agree that it would be good to combine advances in depth estimation methods (such as relative depth map losses or architectural advances), which makes it valuable to discuss these related works.
>
> **“For Waymo dataset, the authors should compare with SOTA self-supervised instance segmentation methods, e.g. SAVi and GroupViT. The paper should also mention the performance of SOTA supervised methods to clarify on the gaps between these two type of methods.”**
>
> We performed a comparison to a fully supervised tracking model similar to Trackformer (Meinhardt et al., 2022), which provides an indication of the gap to fully-supervised methods (supervised result reported for two seeds):
>
>
> |Model             |CoM(%)     | B. mIoU(%)|
> |------------------|-----------|-----------|
> |SAVi++            |4.4 ± 0.0  |50.5 ± 0.3 |
> |Supervised SAVi++ |1.2 ± 0.0  |68 ± 0.3   |
>
> As mentioned in our previous comment (and the general response), we also carried out a comparison to SAVi to measure the improvement of SAVi++ on this dataset as well. We do not consider text-supervision or image datasets in our current benchmarks, which makes GroupViT not suitable for comparison.
>
> **“Can SAVi++'s depth prediction module benefit from the large body of monocular depth estimation paper?”**
>
> Indeed, as mentioned in our reply above, it is likely that SAVi++ can benefit from advances in monocular depth estimation. This is an orthogonal direction to our current contribution, which is why we leave this for future work. We will add a discussion to this point in the paper.
>
> **“It seems that the feature learning can use a strong baseline, e.g. Self-Supervised Representation Learning from Flow Equivariance, ICCV 2021.”**
>
> We do not think that this baseline would contribute much, since our focus is on end-to-end learning without explicit segmentation supervision (as opposed to multi-stage transfer learning using a pretrained backbone and supervised fine-tuning).

---

> > ### Author Response · Authors · 2022-08-02
> > **Response to Reviewer uKMC (continued)**
> >
> > **“The paper does not seem to have stated its limitations. The work has several major limitations, e.g. relying on depth information, far from being competitive with SOTA supervised models, and lacking experimental results on real-world videos.”**
> >
> > In section 4.4 (“Limitations”, p. 9) we state a number of limitations, including those regarding bounding box conditioning in the first frame, temporal consistency, and the gap to existing supervised approaches.
> > Based on your feedback and the comments of the other reviewers, we will substantially expand our discussion of limitations. See our general response above for a summary of additional points we will add to the limitation section.

---

> > ### Comment · Reviewer_uKMC · 2022-08-07
> > **Thank you**
> >
> > I appreciate the authors' effort to address my comments. Specifically, the new results on Waymo comparing with SAVi and Supervised SAVi++ strengthens the paper. The authors are encouraged to address remaining major limitations (e.g. relying on bounding box conditioning in the first frame, lacking results on real-world video datasets) in future work.

---

### Official Review · Reviewer_FUXq · 2022-07-11

**Rating:** 7
**Confidence:** 4
**Soundness:** 3 good
**Presentation:** 4 excellent
**Contribution:** 2 fair

**Summary:**

The paper proposes a model to improve on top of the SAVI model. The key differences include: utilising image depth as a signal for supervision and adopting scaling techniques during training to further boost the performance. The model obtains superior results on both a challenging synthetic dataset (MOVi dataset) and also on real-world driving videos (Waymo Open dataset).


**Questions:**

- Is there any reason why no metrics are reported for the unconditional settings? There are a couple of qualitative comparisons in the paper but no quantitative ones (for e.g. to compare against unconditional SAVi and SIMONe ). I expect the performance to degrade a lot, but I still consider it important to know if this is a scenario where it is worth applying the proposed model.

- Compared to the baselines, the model requires additional supervision (the depth). The author did a great job in explaining how this signal is often available in the real world. However, experimenting with signals coming from predicted depth rather than ground-truth one would be interesting to explore.

- The results reported in Table 1 seem different from the ones reported in [16]. Is the setup in any way different from that one?


**Limitations:**

Since the slots are updated from one step to another, is the model able to “rediscover” an object if it leaves the scene and appears much later? Should we expect the model to attach that object to the original slot, or it will be attached to another random slot? Can this impact the final performance?

**Strengths And Weaknesses:**

Pros:
- With simple modifications, the model obtained clearly superior results compared to the SAVi model.
- The ablation study is detailed and clearly highlights where the improvement comes from
- Obtaining good performance on real-world video is a challenging, missing piece in video object-centric literature, and this paper brings good advancement in this direction.


Cons:
- Even if the model produces strong results, the technical novelty is quite limited. Adding supervision from the depth signal and training techniques such as data augmentation were previously used on video data. However, I agree that clearly isolating the importance of adopting them by the object-centric community is a significant contribution.

---

> ### Author Response · Authors · 2022-08-02
> **Response to Reviewer FUXq**
>
> Thank you for the thorough review and useful feedback. Regarding the remaining questions about our work, please see our detailed point-by-point response below.
>
> **“Is there any reason why no metrics are reported for the unconditional settings?”**
>
> The main reason for this was that our existing metrics require object slots to be matched to ground-truth boxes, which is more difficult to do in the unsupervised case. We have now conducted an evaluation of the unconditional setting using the Hungarian algorithm for matching using otherwise the same CoM metric as in the conditional case. Our results are as follows (lower is better):
>
> |Model          |CoM(%)     |
> |---------------|-----------|
> |SAVi++         |7.8 ± 0.8  |
> |SIMONe + depth |16.7 ± 3.4 |
>
>
> **“Compared to the baselines, the model requires additional supervision (the depth). The author did a great job in explaining how this signal is often available in the real world. However, experimenting with signals coming from predicted depth rather than ground-truth one would be interesting to explore.”**
>
> Thank you for pointing this out. We agree that, while we have opened the door to application domains where depth signals are readily available, there are many real-world settings where depth may be more difficult to measure and one is limited to predicted depth instead.
>
> An experiment with estimated depth is not feasible given the limited rebuttal time. However, we have experimented with “noisy” depth to gain some insights into the performance of SAVi++ when the depth target is suboptimal. We observe that SAVi++ yields good performance even when using additive Gaussian noise with a standard deviation of 40cm added to the depth targets.
>
> |Model               |CoM(%)     | B. mIoU(%)|                   |
> |--------------------|-----------|-----------|-------------------|
> |SAVi++              |4.4 ± 0.0  |50.5 ± 0.3 |3 seeds 500K steps |
> |SAVi++ (noise 10cm) |3.6        |49.89      |1 seed 300K steps  |
> |SAVi++ (noise 40cm) |4.8        |48.35      |1 seed 300K steps  |
>
>
> **“The results reported in Table 1 seem different from the ones reported in [16]. Is the setup in any way different from that one?”**
>
> Indeed, [16] reports results for the SAVi model using a simple CNN encoder. In our paper, we used a more capable baseline for comparison, which is the larger SAVi (ResNet) model variant from Kipf et al. (2022) that uses a ResNet encoder. We will clarify this in the paper.
>
> **“Since the slots are updated from one step to another, is the model able to “rediscover” an object if it leaves the scene and appears much later? Should we expect the model to attach that object to the original slot, or it will be attached to another random slot? Can this impact the final performance?”**
>
> This is an interesting question, but difficult to measure quantitatively in our datasets as many objects that are leaving in Waymo Open usually do not re-appear. We have qualitative evidence that a re-appearing object (such as a car disappearing and reappearing during a turn) is detected later but with a different slot. We have also observed how slots initialized with bounding boxes corresponding to a disappearing object can take on newly appearing objects once the conditioned object has disappeared and it is “free”.
>
> As to how this behavior will impact final performance, it is difficult to say. Clearly, when an object re-appears and it is modeled by a different slot, the quality of the decomposition will be suboptimal in the initial frames of appearance. More generally, there is a lot of headroom to improve the modeling of disappearing and reappearing objects, which is important to address in future work. Two possible approaches are to explicitly model object presence, e.g., as in SQAIR (Kosiorek et al., 2018), or by explicitly attending to past latent states, e.g., as in Slot-VPS (Zhou et al., 2022). We will comment on this in the paper.

---

> > ### Comment · Reviewer_FUXq · 2022-08-08
> > **Response to rebuttal**
> >
> > I thank the authors for properly addressing my concerns. I appreciate the additional metric for the unconditional setting and the effort they made to give some hints about the scenario with noisy depth. I agree that, given the limited rebuttal time, additional experiments using predicting depth are not possible, but it would be interesting to see them in the future. Following the authors response I will increase the score to 7.

---

### Official Review · Reviewer_3HcR · 2022-07-15

**Rating:** 7
**Confidence:** 5
**Soundness:** 4 excellent
**Presentation:** 4 excellent
**Contribution:** 3 good

**Summary:**

SAVi++ is a successor the SAVi, model, which learns to segment objects in video sequences without segment annotations. Both SAVi and SAVi++ take "hints" about where objects might be on the first frame (such as bounding boxes, though they can be trained without such hints.) They both reconstruct optical flow via a bottleneck of object slots.

SAVi showed several key limitations, in particular its poor performance on complex, realistic (or real) datasets and a general inability to segment static objects. SAVi++ makes progress on both of these fronts by using an additional cue / reconstruction target: depth, which may be supplied in addition to or instead of optical flow. Using depth as an additional cue, SAVi++ can learn to segment complex and realistic scenes better than its predecessor, especially static objects. Using sparse depth signals alone from a real-world autonomous driving dataset, SAVi++ can segment many of the objects despite never receiving ground truth segment annotations.

**Questions:**

None

**Limitations:**

Yes

**Strengths And Weaknesses:**

Strengths:

1. This paper makes progress on a challenging problem -- unsupervised segmentation of objects in realistic or real-world videos. This field has been limited mainly to simple synthetic datasets, so pushing toward real scenes is an important contribution.

2. Both key differences between SAVi++ and SAVi are well-motivated: depth is a useful cue for segmentation, and it makes sense that more expressive architectures (and data augmentations) would be important for scaling to real data.

Weaknesses:

1. The description of the literature on human grouping is a little misleading. It's not known whether perceptual grouping in humans is innate or not (L38). Some ability to group moving objects is present as early as two months, and possibly earlier -- it's nearly impossible to measure. What is clear is that grouping ability **matures** from only segmenting moving objects to being able to segment static objects as well. Whether this maturation is a learning process that depends on experience is unknown too, but that much at least seems plausible via a kind of unsupervised learning (and has been recently modeled to some degree in https://openaccess.thecvf.com/content/CVPR2022/html/Bao_Discovering_Objects_That_Can_Move_CVPR_2022_paper.html and https://arxiv.org/abs/2110.06562 and https://arxiv.org/abs/2205.08515.)

2. SAVI's main limitation is that it can't be trained (in its best form) on raw video datasets that contain nothing but RGB frames: it substantially benefits from (and on challenging datasets, effectively requires) object "hints" on the first frame, and also estimates of optical flow. The latter are pretty easy to come by using pretrained models, but many large-scale video datasets do not have bounding boxes for all objects. Moreover, the authors' motivation for requiring these hints (L141-148) is pretty speculative -- I don't know of evidence that infants require people to point at objects in order to group them. Even if there were such evidence, it wouldn't answer the question of how an infant converts the pointing into something like spatial localization of objects. so the initial hints are a practical limitation, and it's likely that humans can get by without them.

SAVI++ seems to still rely on these hints. There are some qualitative results of models trained without hints on the driving dataset, but I don't see quantification on any of them. It's important to know if the hints are still critical for SAVI++'s success -- it is a valuable contribution either way, but doing away with the need for those hints is a challenging problem in its own right.

3. The driving dataset looks like it's very heavy on cars. Being able to segment real scenes is a huge improvement over most earlier unsupervised models, but as far as video datasets go this one probably doesn't have the diversity of objects that others do (e.g. the DAVIS datasets or Kinetics.) The paper would be much stronger if SAVi++ could be shown to work on one of these more diverse video datasets. Short of that, could the authors provide some evaluation of SAVI++'s performance broken down by category on the driving dataset?

4. The reliance on depth signals restricts the use cases of SAVi++ more than SAVi. With optical flow, there are supervised and unsupervised models (e.g. RAFT and SMURF) that transfer well to real-world video datasets, so reliance on flow isn't much of a limitation. But is that true for depth? If the authors could show that sparse GT depth could be replaced with estimated depth (e.g. from stereo cameras, or a transfer from a pretrained monocular depth model) it would substantially relieve this restriction.

---

> ### Author Response · Authors · 2022-08-02
> **Response to Reviewer 3HcR**
>
> Thank you very much for the detailed review and useful feedback. Please find our point-by-point response below.
>
> **“The description of the literature on human grouping is a little misleading. It's not known whether perceptual grouping in humans is innate or not (L38).”**
>
> Thank you for pointing this out. We agree that it is unclear whether perceptual grouping (mainly) develops through experience or is the result of some other kind of maturation process. Hence, we propose to modify the relevant part of line 38 as follows:
>
> "...the ability to organize edges and surfaces into unitary, bounded, and persisting object representations develops through experience and/or maturation ..."
>
> To briefly clarify our understanding of the literature and what led us to write the original sentence in L38 in the first place:
>
> (1) The developmental literature indicates that while maturation may be necessary for certain perceptual abilities to kick in (e.g., binocular perception), it is not sufficient (Kellman & Arterberry, 2000, "The cradle of knowledge: Development of perception in infancy", MIT Press).
>
> (2) It is known that the natural environment provides cues that allow certain grouping principles to be learned (e.g., closure, Kim et al., 2021), and that adults rapidly learn new grouping cues (Zemel et al., 2002).
>
> As you also pointed out, the literature on learned (self-supervised) approaches to object decomposition suggest that some kind of development through experience is certainly a possibility.
>
> **“[SAVi] substantially benefits from (and on challenging datasets, effectively requires) object "hints" on the first frame, and also estimates of optical flow. The latter are pretty easy to come by using pretrained models, but many large-scale video datasets do not have bounding boxes for all objects. Moreover, the authors' motivation for requiring these hints (L141-148) is pretty speculative -- I don't know of evidence that infants require people to point at objects in order to group them. Even if there were such evidence, it wouldn't answer the question of how an infant converts the pointing into something like spatial localization of objects. so the initial hints are a practical limitation, and it's likely that humans can get by without them.”**
>
> Indeed, SAVi benefits from object “hints” in the first frame, which the SAVi authors found was critical to scale from simple synthetic scenes containing simple geometric shapes to more complex synthetic scenes containing everyday real-world objects. In SAVi++, we have adopted this approach as well with the goal of scaling further to real-world visual scenes.
>
> While our main motivation for using object hints in the first frame is thus mainly a pragmatic one, we also felt that this setting shares some similarity to how human visual attention (and how humans parse a visual scene) can be directed via external signals, e.g., by pointing. We acknowledge that our current formulation doesn’t do a good job at communicating this point and may wrongly suggest that infants require people to point at objects to group them. We will update L141-148 accordingly.
>
> We also acknowledge that the use of object hints in the first frame is a practical limitation, which is discussed in the limitations section. Maintaining the performance of SAVi++ without relying on such hints is an exciting direction for future work.
>
> **“There are some qualitative results of models trained without hints on the driving dataset, but I don't see quantification on any of them.”**
>
> Indeed, the main reason for this was that our existing metrics require object slots to be matched to ground-truth boxes, which is more difficult to do in the unsupervised case. To address this, we designed a version of the center of mass (CoM) distance metric that uses Hungarian matching. We obtained the following results (lower is better):
>
> |Model          |CoM(%)     |
> |---------------|-----------|
> |SAVi++         |7.8 ± 0.8  |
> |SIMONe + depth |16.7 ± 3.4 |

---

> > ### Author Response · Authors · 2022-08-02
> > **Response to Reviewer 3HcR (continued)**
> >
> > **“The paper would be much stronger if SAVi++ could be shown to work on one of these more diverse video datasets. Short of that, could the authors provide some evaluation of SAVI++'s performance broken down by category on the driving dataset?”**
> >
> > We agree that the paper would be even stronger if SAVi++ could be shown to work on real-world videos “in the wild”. However, as you also pointed out, with current object-decomposition approaches being limited mainly to simple synthetic datasets, the step that our paper contributes towards real-world scenes is already an important contribution.
> >
> > Compared to real-world videos in the wild, Waymo Open is indeed relatively structured and certainly heavy on cars. DAVIS and Kinetics offer greater complexity in that regard and it is foreseeable that further development of SAVi++ will be needed to truly support these. We will update the limitations section in the paper to reflect this. Having said that, the diversity of Waymo Open videos should also not be underestimated as it concerns scenes recorded outside in the open that include lots of variation in scene backgrounds, clutter, and camera motion. In that sense, we argue that our results are a true step toward end-to-end object-centric learning from real-world videos as the title indicates.
> >
> > Evaluating SAVi++’s performance per category on the driving dataset is a great suggestion. We report mIoU and CoM metrics for the Pedestrian, Car and Cyclist categories below.
> >
> > |               |Car       |Person      |Cyclist      |
> > |---------------|----------|------------|-------------|
> > |Num. instances |15350     |2102        |275          |
> > |CoM(%)         |4.3 ± 0.0 |5.9 ± 0.2   |2.1 ± 0.1    |
> > |B. mIoU (%)    |53.4 ± 0.4|27.2 ± 0.5  |44.5 ± 2.4   |
> >
> > Our results indicate that cars indeed perform best but are closely followed by cyclists. The model performs worse on Pedestrians.
> >
> > **“The reliance on depth signals restricts the use cases of SAVi++ more than SAVi. [...] If the authors could show that sparse GT depth could be replaced with estimated depth (e.g. from stereo cameras, or a transfer from a pretrained monocular depth model) it would substantially relieve this restriction.”**
> >
> > We tend to agree that the reliance on depth signals restricts the use case of SAVi++ more than SAVi. This has been the “cost” of scaling, and we will comment on this in the revised limitations section. At the same time, because of being able to scale, SAVi++ has opened the door to various application domains (such as in robotics) where LiDAR sensors for depth estimation are readily available.
> >
> > An experiment with estimated depth is not feasible given the limited rebuttal time. Instead, we have conducted an experiment with “noisy” depth to provide some indication of the performance of SAVi++ when the depth target is not perfectly accurate (which is likely encountered when using estimated depth). We observe that SAVi++ yields good performance even when using additive Gaussian noise with a standard deviation of 40cm added to the depth targets.
> >
> > |Model               |CoM(%)     | B. mIoU(%)|                   |
> > |--------------------|-----------|-----------|-------------------|
> > |SAVi++              |4.4 ± 0.0  |50.5 ± 0.3 |3 seeds 500K steps |
> > |SAVi++ (noise 10cm) |3.6        |49.89      |1 seed 300K steps  |
> > |SAVi++ (noise 40cm) |4.8        |48.35      |1 seed 300K steps  |

---

### Author Response · Authors · 2022-08-02
**General Response**

We would like to thank the reviewers for their thoughtful comments and positive feedback. We are pleased that the reviewers found that our contribution is significant and makes progress on a challenging problem (Reviewers 3HcR and FUXq), improves on SAVi (Reviewer uKMC), has clear motivation (Reviewer GFoH and 3HcR), has good visualizations (Reviewer GFoH), and has detailed and clear ablations (Reviewer FUXq).

In response to the feedback, we conducted new experiments and we will make changes to the paper to address the points raised by the reviewers. We summarize the main points and experiments below:

**Quantitative metrics on the unconditional models (Reviewers 3HCR, FUXq)**

To quantitatively compare scene decomposition and tracking performance in the unconditional setting, we designed a version of the center of mass (CoM) distance metric that uses Hungarian matching. The matching algorithm finds the optimal assignment of ground-truth bounding box tracks to discovered segmentation mask tracks in order to compute the distance between their respective centers of mass. We obtain the following results (lower is better).

|Model          |CoM(%)     |
|---------------|-----------|
|SAVi++         |7.8 ± 0.8  |
|SIMONe + depth |16.7 ± 3.4 |

**Training with estimated depth (Reviewers 3HCR, FUXq)**

Reviewers suggested investigating estimated depth as a target signal to increase the applicability of SAVi++ to other domains. While we were unable to consider a separate depth prediction model due to the short time frame of the rebuttal, we provide some insight into whether such a direction is likely to succeed. We conducted an experiment to evaluate if an accurate ground-truth depth signal is required for the success of SAVi++ on Waymo Open. We trained SAVi++ models with noisy depth targets instead of ground-truth depth (for which we used additive Gaussian noise with standard deviations of 10cm and 40cm). We found that the model was able to produce good segmentations and object tracks even at the highest considered noise scale of 40cm. See results below.

|Model               |CoM(%)     | B. mIoU(%)|                   |
|--------------------|-----------|-----------|-------------------|
|SAVi++              |4.4 ± 0.0  |50.5 ± 0.3 |3 seeds 500K steps |
|SAVi++ (noise 10cm) |3.6        |49.89      |1 seed 300K steps  |
|SAVi++ (noise 40cm) |4.8        |48.35      |1 seed 300K steps  |

These results indicate that SAVi++ is capable of producing good segmentations and tracking performance despite the inaccurate target depth signal, opening the door for potential utilization of estimated rather than ground-truth depth signals.

**Additional baselines (Reviewer uKMC)**

We evaluated two additional baselines: the original SAVi architecture (Kipf et al., ICLR 2022) and a fully-supervised SAVi++ model variant, which is similar to TrackFormer (Meinhardt et al., CVPR 2022). For the SAVi baseline (conditioned on bounding boxes in the first frame), we experimented with two settings: using RGB targets and using depth targets. We observed that SAVi++ performed significantly better than SAVi (see table below).
For the supervised baseline, we train SAVi++ without the decoder but instead directly supervise the bounding boxes to quantify the gap between SAVi++ and the supervised variant (see table below – *Supervised SAVi++* is reported with 2 seeds only [3rd seed TBD], all other results use 3 seeds).

|Model             |CoM(%)     |B. mIoU(%) |
|------------------|-----------|-----------|
|SAVi (RGB)        |21.5 ± 1.8 |7.9 ± 0.9  |
|SAVi (Depth)      |17.5 ± 5.4 |21.7 ± 8.2 |
|SAVi++            |4.4 ± 0.0  |50.5 ± 0.3 |
|------------------|-----------|-----------|
|Supervised SAVi++ |1.2 ± 0.0  |68 ± 0.3   |


**Evaluation on Waymo Open broken down by category (Reviewer 3HCR)**

In the following table, we break down the SAVi++ results from Table 2, per category, on the Waymo Open dataset. We find that cars dominate the videos but the model is also able to reliably track cyclists even though these are relatively rare.

|               |Car       |Person      |Cyclist      |
|---------------|----------|------------|-------------|
|Num. instances |15350     |2102        |275          |
|CoM(%)         |4.3 ± 0.0 |5.9 ± 0.2   |2.1 ± 0.1    |
|B. mIoU(%)     |53.4 ± 0.4|27.2 ± 0.5  |44.5 ± 2.4   |

---

> ### Author Response · Authors · 2022-08-02
> **General Response (continued)**
>
> **Other real-world datasets (Reviewers 3HCR, uKMC, GFoH)**
>
> Reviewers have noted how our contribution would be even stronger if we provided results on other real-world datasets. The main reason for this was the concern that Waymo Open may not be entirely representative of videos “in the wild”, especially in terms of the diversity of objects. We agree with reviewers that still this is a special domain and other, in the wild, domains may require further work. To clarify this, we will add a discussion to this point to our limitation section.
>
> While we acknowledge the limitations of Waymo Open we also want to emphasize that the diversity of Waymo Open videos should not be underestimated as it concerns scenes recorded outside in the open that include lots of variation in scene backgrounds, clutter, and camera motion. In that sense, we argue that our results on Waymo represent a true step toward end-to-end object-centric learning from real-world videos as the title indicates.
>
> **Improved Limitation Section (Reviewers uKMC, GFoH)**
>
> In response to reviewer feedback, we will expand our limitation section 4.4 to reflect our assumptions about target signals and object hints, model behavior when objects disappear, the gap to supervised methods, the limitations of Waymo Open in comparison to videos in the wild and potential failure modes such as encountered on MOVI-A/B.
>
> For other points and questions raised by the reviewers, please see our response to individual points below each review.

---

> ### Author Response · Authors · 2022-08-09
> **Addendum to General Response**
>
> We have discovered a potential issue with how the CoM distance is computed on Waymo Open in the conditional setting due to how empty segments are evaluated. The CoM metric we report in the paper (unintentionally) assigns a value of 0.0 in this case. We tested another version that assigns a value of 1.0 (corresponding to the maximum distance achievable in a single frame). The result we obtained for SAVi++ in the latter case is 7.2 ± 0.2 %, which is higher than the value of 4.4 % reported in the paper and higher than the value obtained for the BBox Copy baseline.
>
> The choice of 0.0 does not affect our model comparisons and findings as it does not give any of the models a particular advantage, except for the BBox Copy baseline, which does not “predict” empty segments by design. Since it is unclear what the right cost should be for empty segments, this makes this particular comparison under this metric less interpretable.
>
> We propose to measure this behavior using a separate metric, i.e. by measuring in what fraction of the cases an empty segment is predicted even though some part of a box remains visible, which will be included in the paper to complement our current CoM scores. Note that our BBox mIoU metric, which we similarly report for all conditional models, remains unchanged where also the benefit of SAVi++ over the BBox Copy baseline is clearly apparent.

---

### Meta-Review · Area_Chair_rPW6 · 2022-08-23

**Recommendation:** Accept
**Confidence:** Less certain

**Metareview:**

Three out of four reviewers provided positive reviews and scores for this submission. They agreed that SAVI++ makes meaningful improvements over a previously proposed SAVI model. Importantly, while most past approaches evaluate on synthetic data, this submission evaluates the proposed model on a real world dataset. The proposed model clearly improves over the baseline and a clear ablation analysis shows where the improvements come from.

One reviewer had concerns about the evaluation using just one real world dataset. This was also brought up by other reviewers, who mentioned that the Waymo dataset has less diversity and fewer videos than others. While a more thorough evaluation would make this a stronger submission, the leap from synthetic evaluations to real world evaluations in this line of research is notable and sets the bar for future work. I also note, based on the discussion, that the employed dataset is not trivial and has several challenges for the model.

Another concern by the reviewer was about missing baselines. The authors did provide additional baselines in their response. While these baselines do not exactly match the ones requested by the reviewer, I think they provide good evidence that the proposed method is able to employ the depth signal effectively.

Overall, this paper makes solid progress on the problem, provides value to the readers and provides strong results on a real world dataset. Given these reasons, I recommend acceptance.


**Award:**

No

---

### Decision · Program_Chairs · 2022-09-14

Accept